# FINE-TUNING LARGE LANGUAGE MODELS WITH MULTI-AGENT DEBATE SUPERVISION

## ABSTRACT

In this paper, we introduce DebateGPT, a large language model (LLM), which achieves remarkable performance on language generation, comprehension, and reasoning without heavy reliance on resource-intensive human-in-the-loop feedback. DebateGPT is crafted by fine-tuning GPT-3.5 with a limited set of instructions extracted from Alpaca through a novel approach called *multi-agent debate*, achieving comparable performance with GPT-4 in various tasks. We leverage multi-agent debate, harnessing less robust but cost-effective LLMs to generate data without human annotations. Surprisingly, after fine-tuning GPT-3.5 on a modest-size Alpaca dataset obtained by multi-agent debate, DebateGPT shows similar results as GPT-4 on the AlpacaEval test set and showcases remarkable zero-shot generalization to new tasks like commonsense reasoning, factuality and mathematics. For example, DebateGPT outperforms GPT-4 by 2.2% on the arithmetic task. Notably, DebateGPT is much smaller than GPT-4 and only uses a modest dataset. DebateGPT offers an innovative strategy for training highly effective language models without the need for expensive human-in-the-loop feedback or excessively large architectures like GPT-4.

## 1 INTRODUCTION

Recent breakthroughs in large language models (LLMs) like GPT-3.5 and GPT-4 have demonstrated remarkable proficiency in language generation, comprehension, reasoning, and generalization (OpenAI, 2023; Touvron et al., 2023). A growing trend involves fine-tuning these LLMs to enhance specific tasks, such as following user directives or customizing responses for specialized domains. While the human-in-the-loop feedback method (Ouyang et al., 2022) offers a potential avenue for fine-tuning, it can be labor-intensive and time-consuming. An innovative approach is the fine-tuning of LLMs using data from powerhouse models such as GPT-3.5 or GPT-4 (Chiang et al., 2023; Peng et al., 2023).

Utilizing powerful models like GPT-4 guarantees high-quality data but comes with substantial financial implications, costing 20-30 times more than GPT-3.5. While GPT-3.5 is more cost-effective, it occasionally falls short in data quality as noted in (Brown et al., 2020; Ji et al., 2023). This poses an essential dilemma: Is it possible to achieve top-tier data quality without substantial costs or intensive manpower?

In this paper, we introduce a framework that employs the *multi-agent debate* method to generate data for refining pre-trained language models. Recent studies on multi-agent debate (Du et al., 2023) have highlighted its efficacy in enhancing language tasks associated with reasoning, factuality, and mathematics. While the original design of multi-agent debate aimed to bolster text generation during inference, our findings suggest it can also be optimized and repurposed as a data generator for model fine-tuning. Our adaptation of multi-agent debate not only has the potential for high-quality data generation but also offers a more economical alternative to leveraging models like GPT-4.

While multi-agent debate offers potential for high-quality data generation, directly employing existing methods presents challenges. For instance, agents can often converge on incorrect answers without a solid integration mechanism, and the word-length constraints of pre-trained models can limit the use of additional agents or debate rounds. Additionally, responses generated by multi-agent debate can contain unrelated, superfluous information. To address these issues, we introduce three enhancement strategies: 1) Confidence Scoring, which assigns a ranking score to each response

to minimize the influence of weaker replies; 2) Summarization, which streamlines agent responses post each debate round, facilitating the inclusion of more agents or debate rounds and ensuring more accurate results; and 3) Cleaning, which removes irrelevant content from the final debate response. Our findings indicate that these strategies substantially improve the quality of data derived from multi-agent debate.

We then fine-tune pre-trained language models, such as GPT-3.5, using data generated by the improved multi-agent debate. The resulting model, termed DebateGPT, showcases robust zero-shot and few-shot generalization across a wide array of tasks. It not only significantly outpaces GPT-3.5 but also achieves similar performance as GPT-4, despite its smaller model size and being fine-tuned on a mere 5K multi-agent debate data. We provide additional analyses to study the factors that contribute to the effectiveness of DebateGPT. We observed consistent performance enhancements with increasing the number of fine-tuning data, suggesting that, given ample multi-agent debate data, DebateGPT might even eclipse GPT-4's performance.

To summarize, our work has three main contributions:

- We are the first to use multi-agent debate for data generation and model fine-tuning, providing a cost-efficient way to fine-tune LLMs without relying on human annotations. Our method leverages economical language models to generate data through multi-agent debate, ultimately leading to more robust and powerful models.

- We improve the multi-agent debate procedure for data generation, resulting in a significant elevation in data quality. Our method has achieved a 13.2% enhancement in response quality when compared to the original multi-agent debate approach.

- We have developed DebateGPT-3.5, a GPT-3.5 model fine-tuned on 5K examples from the Alpaca dataset with responses generated using our improved multi-agent debate. DebateGPT-3.5 exhibits outstanding performance across a broad spectrum of tasks, including commonsense reasoning, mathematics, factuality, etc. Our evaluations indicate that DebateGPT-3.5 is close to GPT-4 performance in many areas.

These results point to the effectiveness of our proposed method using multi-agent debate data as supervision for language model fine-tuning.

## 2 RELATED WORK

**Training language models.** Language models (LMs), particularly those based on Transformer architectures, have gained significant traction in recent years. Auto-regressive models like GPT (Radford et al., 2018; Brown et al., 2020) predict words in a sequence, with each word being conditioned on its predecessors. On the other hand, models like BERT (Devlin et al., 2018) leverage a masked language modeling approach, where certain words in a sentence are masked out, and the model is trained to predict them. While BERT generally uses random masking approaches, more complex masking approaches, such as dilated sliding window (Beltagy et al., 2020) in the attention mechanism, can better capture long-range dependencies. Furthermore, recent innovations have explored the training of language models using reinforcement learning combined with human feedback (Ziegler et al., 2019). This paradigm has been instrumental in refining the quality of large language models, as evidenced by the enhancements observed in GPT-4 (OpenAI, 2023).

**Fine-tuning language models.** Recent LM fine-tuning (Chiang et al., 2023; Xu et al., 2023; Wang et al., 2023; Gunasekar et al., 2023; Wei et al., 2021; Zhong et al., 2021) methods improve the performance of smaller LMs through the data generated by larger LMs with stronger performance. Recent methods use GPT-3.5 (Brown et al., 2020) and GPT-4 (OpenAI, 2023) to generate responses to natural language instructions (Peng et al., 2023) or conversations (Chiang et al., 2023; Bach et al., 2022), which are used to fine-tune small open-source LMs such as LLaMA (Touvron et al., 2023). Many works (Gunasekar et al., 2023; Peng et al., 2023; Shu et al., 2023; Kaddour et al., 2023) have shown that fine-tuning LMs on smaller yet cleaner datasets yields superior performance compared to training them on larger but noisier datasets. More recent work has used novel prompt methods that improve inference in LLMs such as chain-of-thought prompting and refactored them for data generation (Li et al., 2023; Chung et al., 2022). Inspired by this, we leverage pretrained LMs and

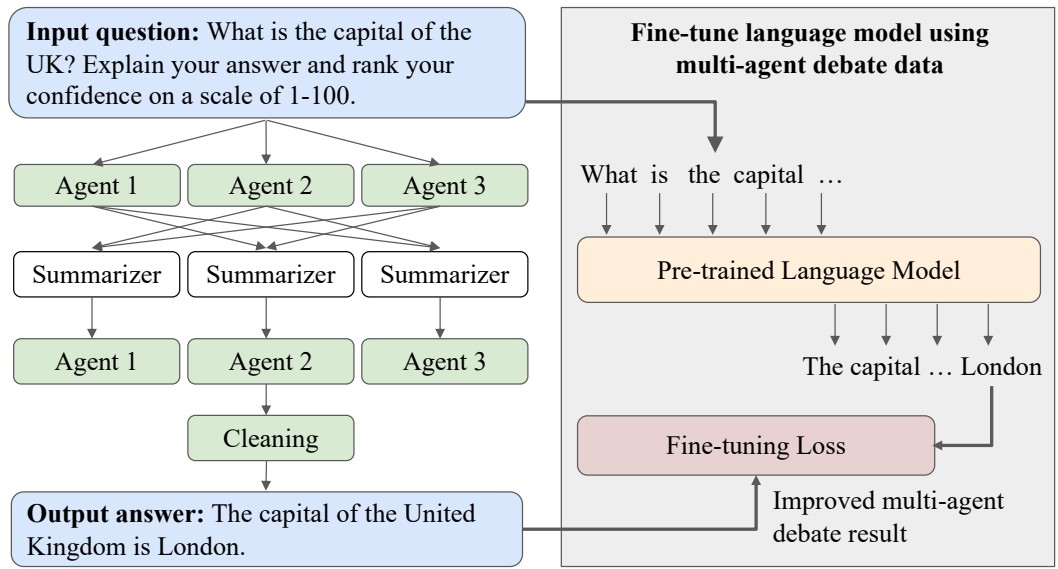

Figure 1: **An overview of DebateGPT.** (Left) We sample 5K instructions from the Alpaca dataset and use a modified version of multi-agent debate to improve generated responses. This involves asking agents for a confidence score, summarizing responses from other agents, and cleaning the final answer. (Right) We collect the question and answer pairs and fine-tune GPT-3.5 using OpenAI FineTuning API.

fine-tune them on a small set of high-quality data and find that they achieve comparable results to GPT-4, which is much larger and trained on vast amounts of data with human feedback.

**Multi-agent debate using language models.** Mutli-agent debate (Du et al., 2023; Liang et al., 2023) is a technique where multiple LMs propose responses and debate with each other, which has been shown to improve zero-shot generalization on tasks related to factuality and reasoning. However, these methods focus on improving text generation ability during interference only, which highly depends on the performance of the original LMs. In this work, we focus on model fine-tuning. We improve the multi-agent debate procedure and use it as a data generator to fine-tune LMs.

## 3 METHOD

In this section. we introduce DebateGPT, a new method for fine-tuning language models using data collected through multi-agent debate. An overview of the proposed method is shown in Fig. 1.

### 3.1 LANGUAGE MODELS FOR TEXT GENERATION

We focus on Transformer-based language models, specifically the GPT family of language models. GPT models use causal attention to predict the next token by only considering previous tokens. The language models are trained to maximize the probability over the text sentence $p(s) = p(w_1) \prod_{i=2}^{T} p(w_i | w_1, \cdots, w_{i-1})$, where $p(w_i | w_1, \cdots, w_{i-1})$ is the conditional probability of token $w_i$ given the previous tokens, and $T$ is the token length of the sentence. The objective during training is to maximize the likelihood of the observed data, which can be achieved using the following loss function:

$$\mathcal{L}(\theta) = -\sum_{i=1}^{T} \log p(w_i | w_1, \cdots, w_{i-1}; \theta), \tag{1}$$

where $\theta$ is the learned model parameters. During training, the model adjusts its parameters (weights and biases) to maximize this likelihood over the training data. This objective ensures that the model learns to generate sequences that are similar to those in its training data.

## 3.2 IMPROVED MULTI-AGENT DEBATE FOR DATA GENERATION

Recent research on multi-agent debate (Du et al., 2023) has shown that multiple language models debate with each other can enhance performance in question-answering tasks that demand reasoning and factual accuracy. While existing multi-agent debate has been employed exclusively during inference to bolster question-answering capabilities, there is an emerging realization of its potential beyond this. Our observations indicate that multi-agent debate can be a prolific source of high-quality data, which can subsequently be harnessed to fine-tune language models. By leveraging the results of debates conducted by weaker language models, we can construct more robust models. This strategy potentially offers a pathway to high-quality training data, circumventing the need for human annotations.

The potential of multi-agent debate is undeniable. However, directly applying the existing methodology for fine-tuning encounters several challenges. First, even though it has been noted that increasing the number of agents and debate rounds can increase the accuracy (Du et al., 2023), the intrinsic constraints on the maximum context window in large language models impede the scalability in terms of both agent count and debate duration. Second, the present configuration of multi-agent debate can also converge to incorrect conclusions, primarily because agents lack an effective mechanism to appropriately assess and integrate the insights and viewpoints of their fellow participants. Finally, the answers produced often carry extraneous debate-centric phrases, like "in the last round" or "the other agents", which don't contribute to model refinement. In light of these constraints, we propose three specific modifications to address each of these challenges, thereby enhancing the efficacy of the multi-agent debate paradigm.

**Summarization:** To handle the constraints on the maximum context window in pre-trained language models, we introduce a summarization model that summarizes responses from other LLM agents per round of debate. Specifically, during the $r$-th round of debate, we first collect the responses of all the other agents from the last round. For agent $a_k$, we send the collected responses $A_k^{r-1} = \{a_1, \cdots a_{k-1}, a_{k+1}, \cdots, a_n\}$ to a summarization model to consolidate these responses and provide a short and clear answer of the given question, where $n$ is the number of agents. In our experiment, we use another GPT-3.5 as the summarizer and keep it distinct from the debating agents. This change allows us to use more agents and debate rounds to obtain more accurate data for fine-tuning downstream models.

**Confidence Scores:** In order to enhance the ability of agents to evaluate and integrate both their own responses and those from other agents effectively, we have integrated a confidence scoring mechanism within the multi-agent debate procedure. Each participating agent is prompted to generate a confidence score, which falls within a range from 1 to 100. This score reflects the agent's level of certainty regarding the accuracy and reliability of its own response. Answers with lower confidence have a smaller influence on the final answer. These confidence scores facilitate a more uniform evaluation while integrating insights from diverse agents.

**Cleaning:** To improve the consistency of the final answer and remove extraneous text generated through multi-agent debate, we introduce a cleaning model that cleans the responses from the last round of debate. We also provide this model with four GPT-4 response examples in the prompt to help it generate text with a concise and coherent format. We use a separate GPT-3.5 as the cleaner, keeping it distinct from the debating agents and the summarization model. The cleaned data is then used for fine-tuning downstream models.

## 3.3 FINE-TUNING LLMS USING MULTI-AGENT DEBATE SUPERVISION

We first randomly sample $5,000$ questions from the Alpaca (Taori et al., 2023) dataset, a dataset consisting of $52,000$ question-answer pairs. We answer the questions using the improved multi-agent debate outlined in Section 3.2. We fine-tune GPT-3.5 using the OpenAI FineTuning API, which supports fine-tunes GPT-3.5 for a user-specified number of epochs. We use the following data format: {*"messages": ["role": "user", "content": "Question", "role": "assistant", "content": "Answer"]*}. The fine-tuned GPT-3.5, *i.e.*, DebateGPT3.5, although considerably smaller compared to GPT-4 and only uses $5,000$ data for fine-tuning, achieved comparable performance with GPT-4 across a diverse range of datasets.

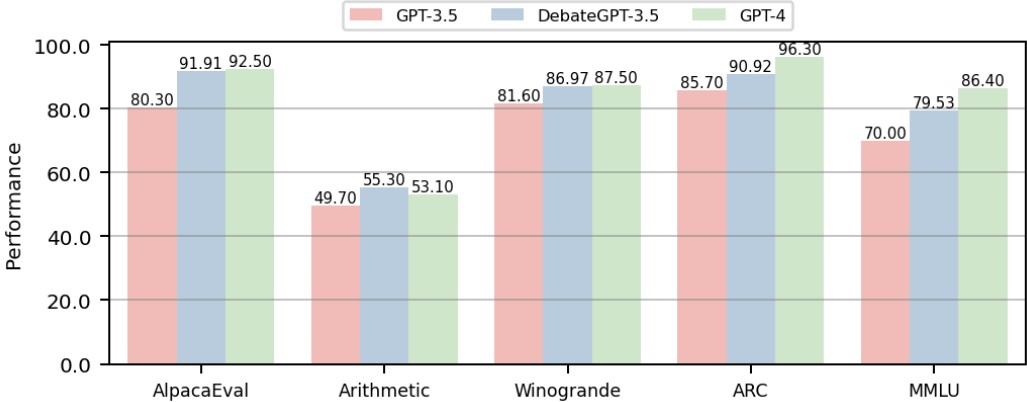

Figure 2: **Results of GPT-3.5, our DebateGPT-3.5, and GPT-4.** We present results comparing the three models on 5 benchmarks related to instruction following, mathematics, commonsense reasoning, and multitask language understanding. DebateGPT-3.5 consistently outperforms GPT-3.5, and is comparable with GPT-4 on several tasks.

# 4 EXPERIMENTS

## 4.1 EVALUATION DATASETS

In this section, we compare the proposed method with the baseline approaches on six datasets.

**AlpacaEval** (Li et al., 2023) consists of 805 examples sampled from the Self-Instruct dataset (Wang et al., 2022), the OpenAssistant dataset (Köpf et al., 2023), ShareGPT (sha, 2023), the Koala test set (Geng et al., 2023), and the HH-RLHF dataset (Bai et al., 2022). Given that a significant portion of the questions in AlpacaEval are open-ended and do not have a single definitive answer, the standard evaluation methodology for this dataset involves employing GPT-4 to rank responses generated by the test model alongside responses generated by `text-davinci-003`. The evaluation then quantifies the percentage of instances where the test model's responses outperform those generated by `text-davinci-003`.

**MMLU** (Hendrycks et al., 2020) consists of multiple-choice questions across 57 subjects related to mathematics, coding, reasoning, and factuality. We evaluate methods on the MMLU test set, which consists of 13,985 multiple-choice questions. Following OpenAI (2023), we provide 5 examples in the prompt. Each example consists of a question, the correct answer choice, and an explanation for the correct answer. We report the accuracy, measuring the number of times the model selects the correct answer.

**Arithmetic** consists of $1,000$ randomly-generated arithmetic problems in the form $a+b*c+d-e*f$. We follow the generation procedure in (Du et al., 2023). Each variable is assigned a random value, constrained to a maximum value of $30$. We generate 1000 examples for evaluation. We compute the percentage of the correct prediction of different models.

**Winogrande** (Sakaguchi et al., 2021) consists of $1,267$ testing questions related to commonsense reasoning. The task involves filling in the blanks in a sentence by choosing the correct option from a pair of options, which are typically proper nouns. Following OpenAI (2023), we provide 5 Winogrande examples in the prompt. We compute the percentage of times the model correctly selects the appropriate option.

**AI2 Reasoning Challenge (ARC)** (Clark et al., 2018) consists of two testing sets: Challenge and Easy. We evaluate different methods on the challenge set which consists of $1,172$ examples of grade-school math problems with multiple choices and science-related questions. We follow OpenAI (2023) and provide 25 examples in the input prompt. We report the percentage of the model generating the correct result.

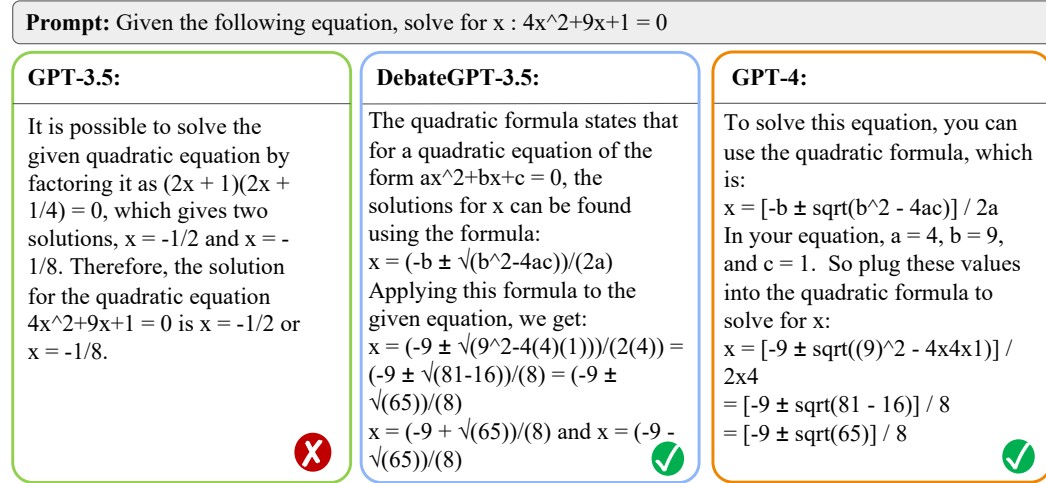

**Prompt:** Given the following equation, solve for x : 4x^2+9x+1 = 0

**GPT-3.5:**

It is possible to solve the given quadratic equation by factoring it as (2x + 1)(2x + 1/4) = 0, which gives two solutions, x = -1/2 and x = -1/8. Therefore, the solution for the quadratic equation 4x^2+9x+1 = 0 is x = -1/2 or x = -1/8.

**DebateGPT-3.5:**

The quadratic formula states that for a quadratic equation of the form ax^2+bx+c = 0, the solutions for x can be found using the formula:
x = (-b ± √(b^2-4ac))/(2a)
Applying this formula to the given equation, we get:
x = (-9 ± √(9^2-4(4)(1)))/(2(4)) = (-9 ± √(81-16))/(8) = (-9 ± √(65))/(8)
x = (-9 + √(65))/(8) and x = (-9 - √(65))/(8)

**GPT-4:**

To solve this equation, you can use the quadratic formula, which is:
x = [-b ± sqrt(b^2 - 4ac)] / 2a
In your equation, a = 4, b = 9, and c = 1. So plug these values into the quadratic formula to solve for x:
x = [-9 ± sqrt((9)^2 - 4x4x1)] / 2x4
= [-9 ± sqrt(81 - 16)] / 8
= [-9 ± sqrt(65)] / 8

Figure 3: **An example from the AlpacaEval dataset.** We show that DebateGPT-3.5 can reason about mathematics problems, in this case solving a problem using the quadratic formula. Both DebateGPT-3.5 and GPT-4 get the problem correct while GPT-3.5 identifies the wrong solution. The generated text is truncated to save space in this example.

## 4.2 BASELINES

This section focuses on comparisons of DebateGPT-3.5, GPT-3.5, and GPT-4. We reported comprehensive comparisons with more methods in Appendix A.

**GPT3.5** is an autoregressive language model built on top of `text-davinci-002`. The `davinci` model series consists of transformer decoder architectures with 175B parameters[1]. GPT-3.5 was formed through optimizing `text-davinci-002` for code generation and chatting capabilities. The specific model architecture and training dataset are currently unreleased by OpenAI.

**GPT4** is the newest LLM from OpenAI (OpenAI, 2023) and has set the state-of-the-art for many benchmarks related to language generation and understanding, code generation, reasoning, and factuality. While the specific architecture and dataset details are unknown and have not been confirmed, the current understanding is that GPT-4 has the order of magnitude of one trillion parameters, making it 10 times larger than GPT-3 and potentially ten times larger than GPT-3.5. GPT-4 is fine-tuned using RLHF (Ziegler et al., 2019) techniques, where human annotators judge and score generated outputs from the model to improve model performance and prevent toxic or biased outputs.

For the MMLU, ARC, and Winogrande datasets, we rely on the GPT-3.5 and GPT-4 results as reported in (OpenAI, 2023). As for the AlpacaEval and Arithmetic datasets, we obtain results by utilizing the `gpt-3.5-turbo-0613` and `gpt-4-0613` models.

## 4.3 COMPARSION WITH BASELINES

We show the performance of DebateGPT-3.5 in comparison to baselines in Fig. 2. Our DebateGPT-3.5 is GPT-3.5 fine-tuning on data generated using the improved multi-agent debate (4 agents and 3 debate rounds). Our results show that DebateGPT-3.5 improves by 5-11% over GPT-3.5 across different datasets. Most excitingly, we see that DebateGPT achieves comparable performance to GPT-4, even surpassing GPT-4 on the Arithmetic dataset. Our DebateGPT-3.5 only uses $5,000$ Alpaca data for fine-tuning, but it can generalize to new datasets, such as AlpacaEval, which is a combination of data from multiple resources. DebateGPT-3.5 can also solve math problems and tasks requiring commonsense reasoning.

We see that improvements are not evenly observed, specifically factuality datasets improve marginally over GPT-3.5 and don't completely achieve GPT-4 level performance. One potential reason for this result is the small fine-tuning dataset that may need more examples related to factual-

---

[1]`https://platform.openai.com/docs/model-index-for-researchers`

**Prompt:** Can you answer the following as accurately as possible? Which of the following changes to a double-slit interference experiment would increase the width of the fringes? A) Use of light of a shorter wavelength B) Move the screen closer to the slits C) Move slits closer D) Use light with lower save speed. Explain your answer.

**GPT-3.5:**

The correct answer is A).

Using light of a shorter wavelength will increases the width of the interference fringes. This is because the formula for the angular width of the interference fringes in a double-slit experiment

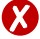

**DebateGPT-3.5:**

C) Move slits closer. Moving the slits closer to each other will actually increase the width of the interference fringes. This is because, according to the formula for angular width, if you decrease the distance between the slits (d), the linear width of the fringes (w) increases. So, this choice would indeed increase the width of the fringes.

✅

**GPT-4:**

Choice C) Move the slits closer together would lead to smaller fringes, not larger. Choice A and Choice D would both result in smaller fringes. Choice B) Move the screen closer to the slits would decrease the fringe width. None of the answer choices will increase the width of the fringes. A change that would increase the fringe width would be to move the screen farther from the slits, not closer.

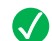

Figure 4: **An example from the MMLU dataset.** We show that DebateGPT-3.5 has improved reasoning about science and physics problems. GPT-3.5 identifies the wrong answer and GPT-4 eliminates all answers, incorrectly associating C) with decreasing width. The generated text is truncated to save space in this example.

ity and reasoning to achieve stronger performance. For example, DebateGPT-3.5 improves on many topics in the MMLU dataset but does not improve on law-based questions since such instructions do not seem to occur in the Alpaca dataset and therefore, do not appear in our fine-tuning dataset. Furthermore, DebateGPT-3.5 does not improve on questions related to specific historical events, questions that GPT-4 systematically gets correct due to a much larger training set and potentially larger knowledge base.

## 4.4 QUALITATIVE RESULTS

In Fig. 3 and Fig. 4, we show example results of different methods. Fig. 3 is a mathematics problem associated with using the quadratic formula from the Alpaca dataset. While GPT-3.5 fails by attempting to use factorization, DebateGPT-3.5 and GPT-4 correctly apply the quadratic formula and arrive at the correct result. This shows the improved mathematical reasoning ability of DebateGPT-3.5 over the original GPT-3.5. In Fig. 4, we show an example from the MMLU dataset related to light and wave processing in physics. Both GPT-3.5 and GPT-4 fail at this problem. GPT-4 cannot identify a specific answer, and GPT-3.5 selects the wrong answer. DebateGPT-3.5 improved the factuality and reasoning ability by fine-tuning GPT-3.5 using multi-agent debate data and can generate the answer correctly. We show more examples in Appendix B.

## 5 ANALYSIS

In this section, we investigate factors contributing to DebateGPT-3.5's ability to achieve performance comparable to that of GPT-4, despite its significantly smaller model size. We hypothesize two possible factors to explain the effectiveness of DebateGPT-3.5.

### 5.1 HOW MUCH DOES DATA QUALITY MATTER?

We first hypothesize that the improved multi-agent can generate higher-quality data which are useful for model fine-tuning. In Table 1, we compared the data generated by GPT-3.5, the original multi-agent debate using GPT-3.5 proposed by (Du et al., 2023), our improved multi-agent debate, and GPT-4. Both the original multi-agent debate and our improved version use 4 agents and 3 debate rounds. We randomly sample 500 examples from the Alpaca dataset, excluding the ones used for model fine-tuning. As many questions are open-ended, we follow the evaluation setting used in

| Model | Win Rate (%) ↑ |
|---|---|
| GPT3.5 | 72.4 |
| Multi-agent Debate (GPT3.5) (Du et al., 2023) | 75.2 |
| Multi-agent Debate (GPT3.5) + Summarization | 78.8 |
| Multi-agent Debate (GPT3.5) + Confidence | 79.6 |
| Multi-agent Debate (GPT3.5) + Confidence + Summarization | 82.0 |
| Multi-agent Debate (GPT3.5) + Confidence + Summarization + Cleaning (Ours) | 88.6 |
| GPT-4 | 89.0 |

Table 1: **Results of improved multi-agent debate.** We sample 500 examples from the Alpaca dataset and generate responses using GPT-3.5, the multi-agent debate proposed in (Du et al., 2023), our improved multi-agent debate, and GPT-4. Incorporating the summarization step, confidence scoring, and the cleaning step can significantly improve the multi-agent debate performance. Our method shows an improvement of 13.4% compared to the original multi-agent debate method, bringing it to a level comparable with GPT-4.

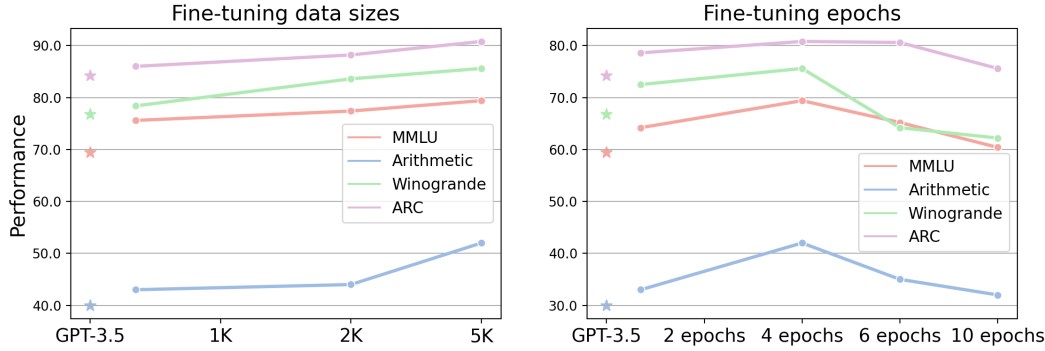

Figure 5: **DebateGPT-3.5 performance across varied Data (left) and fine-tuning epochs (right).** Increasing the number of data consistently improves performance. Fine-tuning GPT-3.5 with 1K debate data already outperforms GPT-3.5. Using 4 epochs leads to the best performance.

AlpacaEval that uses GPT-4 to rank the answer generated by a method and the answer generated by `text-davinci-003`. The win rate over `text-davinci-003` are reported.

Firstly, it is evident that employing a multi-agent debate approach yields superior results compared to GPT-3.5. We find that incorporating a summarization step after each debate round leads to a notable performance enhancement, amounting to a 3.6% improvement over the original multi-agent debate method (Du et al., 2023). Furthermore, the introduction of a ranking score that reflects the confidence in the generated result can further elevate performance by an additional 0.8%. When both summarization and the confidence score are used in tandem, there is a substantial increase (6.8%) in the win rate. Our DebateGPT-3.5, which combines the features of summarization, confidence scoring, and cleaning, surpasses the original multi-agent debate method by 13.4%, which is comparable with GPT-4 results.

## 5.2 WHAT FACTORS ARE IMPORTANT FOR FINE-TUNING RESULTS?

We then put forward the hypothesis that the quantity of data plays a significant role in influencing the results of the fine-tuning process. In Fig. 5 (left), we assess models that have undergone fine-tuning using datasets containing 1K, 2K, and 5K samples extracted from Alpaca, with answers generated using the improved multi-agent debate method. For evaluation purposes, we randomly select 500 examples from each dataset.

Remarkably, even with just 1K data for fine-tuning (64.6%), we observe performance that surpasses GPT-3.5 (55.2%) on the MMLU dataset. We have the same conclusion on other datasets. This substantial improvement underscores the high quality of the data generated through the multi-agent debate approach. Furthermore, we find that using a larger volume of data consistently leads to

improved results across all datasets. This suggests that there is room for further enhancement of DebateGPT-3.5 with an increase in the amount of data used for fine-tuning, increasing its chances of surpassing GPT-4.

We also posit that the fine-tuning strategy plays a pivotal role in shaping the ultimate results. However, due to constraints imposed by the OpenAI Fine-tuning API, our ability to experiment with fine-tuning strategies is limited to testing the influence of fine-tuning epochs. In Fig. 5 (right), we present the performance of models fine-tuned with varying numbers of epochs. Intriguingly, we observe that the optimal performance is achieved when using 4 epochs. Going beyond this threshold results in the emergence of overfitting issues. For a more detailed view of the fine-tuned models, please refer to the perplexity plot in Appendix C.

### 5.3 WHY NOT GPT4 DIRECTLY?

Although the improvements seen in DebateGPT-3.5 are impressive, we could have theoretically achieved these results by fine-tuning the model on Alpaca data with responses generated by GPT-4 instead of multi-agent debate. In Fig. 6, we show that generating data using our multi-agent debate (*Debate (ours)*) is cheaper than GPT-4 based on the current costs from OpenAI. *Debate (no sum)* means the cost of multi-agent debate without using the final cleaning step.

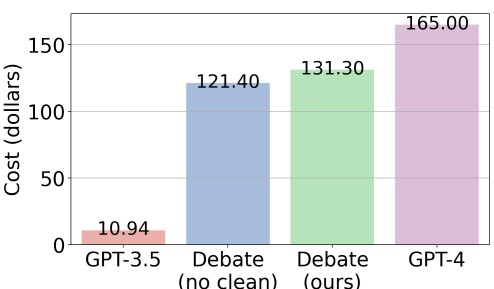

Figure 6: **Costs of generating 500 Alpaca data using GPT-3.5, multi-agent debate, and GPT-4.** Our current setting of multi-agent debate (*Debate (ours)*) is cheaper than using GPT-4.

Furthermore, under the current OpenAI pricing, it costs $0.008 per epoch to fine-tune a dataset with 1K tokens. Using the OpenAI `tiktoken` package, we calculated that our current dataset of 5K Alpaca examples consists of 1.3M tokens. Therefore, fine-tuning on our dataset costs only $10 per epoch, and therefore, our total fine-tuning cost ranged from $20-$100, a neglible cost compared to using GPT-4. Additionally, the current OpenAI pricing makes using fine-tuned models like DebateGPT-3.5 to generate text half as expensive using GPT-4 [2].

## 6 CONCLUSION

We introduced a new method for fine-tuning LLMs through data derived from multi-agent debate. This resulted in DebateGPT, an LLM based on GPT-3.5, fine-tuned using responses to 5,000 examples randomly sampled from the Alpaca dataset. The responses to these examples were enhanced by multi-agent debate techniques coupled with the introduction of confidence scores, summarization, and cleaning. This approach not only elevates the performance of DebateGPT beyond GPT-3.5 but also achieves results on par with GPT-4 across six datasets spanning instruction-following, factuality, commonsense reasoning, and mathematics.

Our method has demonstrated results that are comparable to GPT-4 across a range of tasks. However, it is crucial to note that we haven't yet assessed its performance on an exhaustive list of tasks, so making a definitive comparison might be premature. What's particularly exciting about our approach is that it offers an innovative paradigm for training language models without the need for human-annotated data. This suggests that it is possible to achieve impressive outcomes with smaller models trained on limited but high-quality data, challenging the conventional belief that larger models necessitate vast datasets with human feedback. One avenue to enhance our model's capabilities lies in refining the multi-agent debate process. As we move forward, we intend to delve deeper into this aspect with the hope of further elevating the quality of our fine-tuned models. Furthermore, our proposed approach can be seamlessly integrated with open-source language models like Llama-2 (Touvron et al., 2023) at no extra expense. Due to computational constraints, these results weren't showcased. Nonetheless, we plan to incorporate and release this model in the future.

---

[2]https://openai.com/pricing

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

# Appendix

In this appendix, we provide additional comparison results with more models in Appendix A. We show additional qualitative examples in Appendix B. The perplexity values over epochs of fine-tuning GPT-3.5 are shown in Appendix C.

## A    COMPARISONS WITH MORE MODELS

We present further comparisons between our proposed method and the state-of-the-art methods on the AlpacaEval dataset in Table 2, MMLU in Table 3, ARC in Table 4, and Winogrande in Table 5. We see that DebateGPT-3.5 is comparable with and even surpasses several state-of-the-art models.

## B    QUALITATIVE EXAMPLES

We include more examples of DebateGPT-3.5 outputs in comparison with GPT-3.5 and GPT-4. Fig. 7, Fig. 8, Fig. 9, Fig. 10, Fig. 11 show the successful examples. Our findings consistently demonstrate superior performance of DebateGPT-3.5 over GPT-3.5 across all datasets. Moreover, there are numerous instances where DebateGPT-3.5 accurately answers questions that GPT-4 misses.

We include two failure examples of DebateGPT-3.5 in Fig. 13 and Fig. 12. In Fig. 12, we see that model halluncinations in GPT-3.5 remain in DebateGPT-3.5, despite fine-tuning. References to philosophical works rarely occur in the Alpaca dataset and likely, questions related to social sciences and historical references are likely to have improved very little. Similarly, in Fig. 13, we see example of incorrect mathematical calculations. While mathematical processing has improved across a majority of datasets, we see that DebateGPT-3.5 is still susceptible to incorrect mathematical calculations in the AlpacaEval set.

## C    PERPLEXITY CURVES DURING FINE-TUNING

We show fine-tuning perplexity curves while fine-tuning GPT-3.5 for up to 10 epochs in Fig. 14. These perplexity curves represent how well the model fits to the fine-tuning dataset by providing a metric for how well the model captures the output token probability distribution. Higher perplexity indicates a worse fit on the fine-tuning dataset. We see that for all dataset sizes, perplexity decreases, indicating a strong fit to our data. In general, we can use this curve in tandem with Fig. 5 to see that the GPT-3.5 fine-tuning can also overfit as DebateGPT-3.5 versions that are fine-tuned for 10 epochs see a significant decrease in performance across datasets that need generalization. We find that the best model setting occurs at 4 epochs of training, associated with 600 steps in Fig. 14.

## D    CONFIDENCE SCORING

In Table 8, we analyze the confidence score assigned by the LLMs during multi-agent debate. For 100 examples from our 5K fine-tuning dataset, we assess whether the confidence score assigned in the first round and last round of debate accurately reflects the correctness or quality of the response. For this anaLysis, we assign a yes/no label and count report the percentage of yes labels in our assessment. We find that the last round of debate improves the confidence scoring over the first round by close to 20%. Furthermore, we find that for the 18 examples that we believed to have unfaithful confidence scoring, 10 of them were due to the agent assigning a lower confidence score despite having a higher quality/correct answer.

## E    SUMMARIZATION

In Table 7, we used different summarization prompts to test the best method for consolidating information from other agents during multi-agent debate. We found that the summarization prompt design played a considerable role in generating high quality data. Empirically, we observed that the best summarization prompt asked to consolidate information from other agents into a single response

as this better balanced responses between agents and prevented information loss from summarizing the response of each agent, which was longer and more inaccurate. We found that too much consolidation also had an effect on performance, where specifying to consolidate to one sentence seemed to cause information loss.

## F  DATA GENERATION BASELINES

We compare our improved multi-agent debate method with other methods of generating responses to the Alpaca dataset. We report our findings in Table 6. We see that compared to other prompting methods used to automatically generate data such as chain-of-thought-prompting (Li et al., 2023), our method shows an improvement in response quality. This strengthens the argument for using multi-agent debate for data generation with our proposed changes. We note that chain-of-thought prompting generates answers of similar quality to the originally proposed version of multi-agent debate from Du et al. (2023). This further justifies that our changes were meaningful for separating the response quality from our modified multi-agent debate from other methods.

## G  DEBATE FINE-TUNING DATASET CONSTRUCTION

In constructing our 5K fine-tuning dataset, we could have included the entire debate demonstration instead fine-tuning on the response from the final round, potentially leading to improvements in the evaluation performance. We compare fine-tuning with the entire debate demonstration to fine-tuning with the final round response to construct DebateGPT-3.5 in Table 9. We see that using the final response consistently improves performance on 100 MMLU examples.

| Model | Win Rate (%) ↑ |
|---|---|
| GPT-4 (*the version used in the leaderboard*) | 95.28 |
| LLaMA2 Chat (70B) | 92.66 |
| Claude 2 | 91.36 |
| ChatGPT | 89.37 |
| Vicuna 33B v1.3 | 88.99 |
| Claude | 88.39 |
| Vicuna 13B v1.3 | 82.11 |
| GPT-3.5 (*the version used in the leaderboard*) | 81.71 |
| LLaMA2 Chat 13B | 81.09 |
| Vicuna 7B v1.3 | 76.84 |
| WizardLM 13B | 75.31 |
| Guanaco 65B | 71.80 |
| LLaMA2 Chat 7B | 71.37 |
| Vicuna 13B | 70.43 |
| LLaMA 33B OASST RLHF | 66.52 |
| Guanaco 33B | 52.61 |
| Davinci003 | 50.00 |
| Guanaco 7B | 46.58 |
| Falcon 40B Instruct | 45.71 |
| Alpaca Farm PPO Sim (GPT-4) 7B | 41.24 |
| Alpaca 7B | 26.46 |
| Davinci001 | 15.17 |
| GPT-3.5 (`gpt-3.5-turbo-0613`) | 80.30 |
| DebateGPT-3.5 (`gpt-3.5-turbo-0613`) (Ours) | 91.91 |
| GPT-4 (`gpt-4-0613`) | 92.50 |

Table 2: **Comparisons of our DebateGPT-3.5 and the state-of-the-art methods on AlpacaEval.** The results are collected from the public leaderboard. We test the stable version of GPT-3.5 (`gpt-3.5-turbo-0613`) and GPT-4 (`gpt-4-0613`) in our experiments. The results obtained from GPT-3.5 and GPT-4 may differ from those reported on the leaderboard due to variations in model versions. Our DebateGPT-3.5 is much smaller than the best models in the leaderboard but achieves comparable performance.

| Model | MMLU Accuracy (%) ↑ |
|---|---|
| GPT-4 | 86.40 |
| Codex + REPLUG LSR | 71.80 |
| Codex + REPLUG | 71.40 |
| GPT-3.5 | 70.00 |
| U-PaLM | 70.70 |
| PaLM (540B) | 69.30 |
| Codex | 68.30 |
| Chinchilla | 67.50 |
| LLaMA (65B) | 63.40 |
| Gopher | 60.00 |
| GAL 120B | 52.60 |
| GPT-3 175B | 43.90 |
| GPT-NeoX-20B | 33.60 |
| Gopher-7.1B | 29.50 |
| Gopher-1.4B | 27.30 |
| GPT-3 13B | 26.00 |
| GPT-3 2.7B | 25.90 |
| Gopher-0.4B | 25.70 |
| DebateGPT-3.5 (`gpt-3.5-turbo-0613`) (Ours) | 72.13 |

Table 3: **Comparison of our DebateGPT-3.5 and state-of-the methods on MMLU.** MMLU accuracy leaderboard with 5-shot prompting. We evaluate DebateGPT by sampling 5 questions from the MMLU training set and providing answers and explanations in the prompt. We access the rest from the public leaderboard. DebateGPT-3.5 is comparable with the state-of-art models on the MMLU dataset.

| Model | ARC Accuracy (%) ↑ |
|---|---|
| GPT-4 | 96.30 |
| ST-MoE-32B | 86.52 |
| GPT-3.5 | 85.70 |
| UnifiedQA+ARC MC/DA + IR | 81.40 |
| UnifiedQA - v2 (T5-11B) | 81.14 |
| GenMC | 79.86 |
| ZeroQA | 78.58 |
| UnifiedQA (T5-11B; finetuned) | 78.50 |
| CGR + AristoRoBERTav7 | 68.94 |
| AMR-SG+AristoRoBERTaV7 | 67.75 |
| arcRoberta | 67.15 |
| xlnet+roberta | 67.06 |
| ARCCorpusRoBERTa | 66.89 |
| AristoRoBERTaV7 | 66.47 |
| Attentive Ranker (ALBERT) | 62.97 |
| ARChaeopteryx | 62.46 |
| Multi-Task BERT | 60.58 |
| AristoBERTv7 | 57.76 |
| UnifiedQA (BART-uncased-large) | 54.95 |
| QA Transfer | 53.84 |
| Multi-Task BERT (Single Model) | 48.29 |
| Attentive Ranker (BERT) | 44.71 |
| BERT MRC Transfer (Single Model) | 44.62 |
| DebateGPT-3.5 (`gpt-3.5-turbo-0613`)(Ours) | 87.89 |

Table 4: **Comparison of our DebateGPT-3.5 and the state-of-the-art models on ARC.** ARC accuracy with 25-shot prompting. 25 examples are sampled from the ARC training set and provided to DebateGPT-3.5 with gold standard answers and explanations. The rest of the results are collected from a public leaderboard. We show that DebateGPT-3.5 achieves comparable and better performance than the models on the leaderboard.

| Model | WinoGrande Accuracy (%) ↑ |
|---|---|
| GPT-4 | 87.50 |
| PaLM 2-L | 83.00 |
| GPT-3.5 | 81.60 |
| PaLM 540B | 81.10 |
| PaLM 2-M | 79.20 |
| PaLM 2-S | 77.90 |
| PaLM 62B | 77.00 |
| LLaMA 65B | 77.00 |
| LLaMA 33B | 76.00 |
| Chinchilla 70B | 74.90 |
| phi-1.5-web-1.3B | 74.00 |
| LLaMA 13B | 73.00 |
| GPT-3 175B | 70.20 |
| LLaMA 7B | 70.10 |
| Gopher 280B | 70.10 |
| DebateGPT-3.5 (`gpt-3.5-turbo-0613`) (Ours) | 87.00 |

Table 5: **Comparisons of our DebateGPT-3.5 and the state-of-the-art methods on WinoGrande** Wino-grande accuracy leaderboard with 5-shot prompting. We evaluate DebateGPT-3.5 by sampling 5 questions from the WinoGrande training set and providing gold-standard answers and explanations in the prompt. We see that DebateGPT-3.5 is out-performs all other models other than GPT-4.

| Model | Win Rate (%) ↑ |
|---|---|
| GPT3.5 | 74.2 |
| Chain of Thought Prompting | 78.4 |
| Multi-agent Debate (GPT3.5) (Du et al., 2023) | 77.2 |
| Multi-agent Debate (GPT3.5) + Confidence + Summarization + Cleaning (Ours) | 81.2 |

Table 6: **Data Generation Baseline Comparison** We sample 1000 fine-tuning examples from the 5K Alpaca examples and generate responses using GPT-3.5, chain-of-thought prompting with GPT-3.5, the multi-agent debate proposed in (Du et al., 2023), and our improved multi-agent debate. Incorporating the summarization step, confidence scoring, and the cleaning step can significantly improve the multi-agent debate performance and improve performance over other debate methods. Our method shows a 2.9% improvement in responses quality over these 1000 examples.

| Summarization Prompt | Win Rate (%) ↑ |
|---|---|
| Summarize the responses from different agents into a short sentence. | 65 |
| Write a summary of the different solutions from each of the individual agents and briefly explain their reasoning. | 75 |
| Please summarize the responses from different agents by consolidating the responses from the agents into one response for the given question (Ours). | 79 |

Table 7: **Summarization Prompts**: We used different prompts to perform summarization during multi-agent debate, varying the different vocabulary such as *summarize* versus *consolidate*. We found that the usage had an effect on response quality as the first prompt decreased the number of words in the summary did not appropriately balance the responses from multiple agents.

**Prompt:** Find the equation of the line which passes through the given point and is parallel to the given vector. Point: (2, 1), Vector: (−2, 1)

**GPT-3.5:**

First, we can use the direction vector of the given vector, which is ⟨-2, 1⟩, to determine the direction of the line we are looking for. Next, using the point-slope formula, we can write the equation of the line in slope-intercept form: y - y1 = m(x - x1),where (x1, y1) is the given point and m is the slope of the line. Simplifying the equation, we have:
y = -1/2x + 2 - 1/2,
y = -1/2x + 3/2. 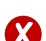

**DebateGPT-3.5:**

To find the equation of the line parallel to the given vector passing through the given point, we can use the point-slope form of the equation of a line. The slope of this line is the same as the slope of the given vector, i.e., -1/2. Plugging in the coordinates of the given point (2, 1) and the slope into the point-slope form, we have:
y - 1 = (-1/2)(x - 2)
y - 1 = -1/2x + 1:
y = -1/2x + 2
Therefore, the equation is y = -1/2x + 2. ✓

**GPT-4:**

In two dimensions, the equation of a line can be written in slope-intercept form: y = mx + b, where m is the slope and b is the y-intercept.
In this case, the given vector is (−2, 1), so the slope of the line m is 1/-2 = -1/2. Since we know the line passes through the point (2, 1), we can substitute these values into the equation and solve for b (the y-intercept):  1 = (-1/2)x2 + b,
1 = -1 + b,
b = 2.
So, the equation of the line is y = -1/2x + 2. 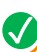

Figure 7: **AlpacaEval Example** AlpacaEval example where DebateGPT-3.5 and GPT-4 exhibit similar quality outputs and arrive at the correct answer. GPT-3.5 obtains the wrong result due to finding the incorrect y-intercept.

**Prompt:** Can you answer the following question as accurately as possible? Controlling for inflation and PPP-adjustment, about how much did GDP per capita increase from 1950 to 2016 in Japan?: A) by 5 fold, B) by 10 fold, C) by 15 fold, D) by 20 fold

**GPT-3.5:**

Based on available data, the GDP per capita in Japan increased significantly from 1950 to 2016. However, without specific information on the exact inflation and PPP-adjustment figures, it is difficult to provide an exact amount of increase. That being said, a conservative estimate would be an increase of about 5 fold (A). 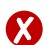

**DebateGPT-3.5:**

It is clear that controlling for inflation and PPP-adjustment, GDP per capita in Japan increased significantly from 1950 to 2016. The consensus is that GDP per capita in Japan increased by around 15 fold during this period. Therefore, the most accurate answer to the question would be C) by 15 fold. 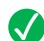

**GPT-4:**

According to data from the World Bank, Japan's GDP per capita (controlling for inflation and PPP-adjustment) has increased significantly from 1950 to 2016. Specifically, in 1950, Japan's GDP per capita was $1,926 (in 2011 international $). By 2016, it had risen to $38,895 (in 2011 international $). That's an increase of approximately 20 times, or 20 fold. (D) ✗

Figure 8: **MMLU example** DebateGPT-3.5 seems to have improved performance related to world knowledge and current events, correctly accessing records related to PPP of Japan in 1950. While GPT-4 has the same capabilities, it seems to access the wrong information for PPP of Japan in 1950.

**Prompt:** Can you fill in the blank in the following sentence using one of the two given options? At night, Jeffrey always stays up later than Hunter to watch TV because _ wakes up late. Option 1:Jeffrey Option 2:Hunter.Give your answer as a number, so 1 for option 1 or 2 for option 2 at the end of your response.

| GPT-3.5: | DebateGPT-3.5: | GPT-4: |
|---|---|---|
| Option 2: Hunter. While either option can work in this case, Hunter makes more logical sense and reads better in comparison to Jeffrey. My answer is Option 2. | Jeffrey wakes up late. (Option 1). Jeffrey staying up to watch TV late at night implies that this causes Jeffrey to wake up late in the morning. Therefore, the correct option is (1). | The correct sentence would be: At night, Jeffrey always stays up later than Hunter to watch TV because Jeffrey wakes up late. So, the answer is 1. |

Figure 9: **Winogrande example**. DebateGPT-3.5 has improved reference resolution capabilities, compared with GPT-3.5. We notice that GPT-3.5 seems to use proximity as a factor in co-reference resolution, which DebateGPT improves on.

**Prompt:** Can you answer the following question as accurately as possible? A group of engineers wanted to know how different building designs would respond during an earthquake. They made several models of buildings and tested each for its ability to withstand earthquake conditions. Which will most likely result from testing different building designs? (A) buildings will be built faster (B) buildings will be made safer (C) building designs will look nicer (D) building materials will be cheaper. Explain your answer.

| GPT-3.5: | DebateGPT-3.5: | GPT-4: |
|---|---|---|
| The answer is (None of the above). None of the answers adequately address how the group of engineers will respond during an earthquake. Therefore, my answer is (None of the above). | Based on the careful analysis of the language used in the question, the use of \"ability to withstand earthquake conditions\" and the purpose of testing building models, it can be concluded that the most likely outcome of the engineering group's testing of different building designs will be to create safer buildings, as the focus is on improving their ability to withstand earthquakes. Therefore, the answer to the question is (B) buildings will be made safer | The correct answer is (B) buildings will be made safer. Explanation: When engineers test various building designs for their ability to withstand earthquake conditions, their primary objective is to understand the structural performance and safety of each design under such extreme conditions. The results from these tests would ideally inform design improvements to ensure that buildings are better equipped to survive and protect occupants during earthquakes. |

Figure 10: **ARC Example**. DebateGPT-3.5 expresses general reasoning about engineering and design goals to handle and solve certain problems, similar to GPT-4. Instruction-tuning seems to improve reasoning capabilities.

**Prompt:** Find the equation of the line which passes through the given point and is parallel to the given vector. Point: (2, 1), Vector: (−2, 1)

| GPT-3.5: | DebateGPT-3.5: | GPT-4: |
|---|---|---|
| First, we can use the direction vector of the given vector, which is ⟨-2, 1⟩, to determine the direction of the line we are looking for. Next, using the point-slope formula, we can write the equation of the line in slope-intercept form: y - y1 = m(x - x1), where (x1, y1) is the given point and m is the slope of the line. Simplifying the equation, we have: y = -1/2x + 2 - 1/2, y = -1/2x + 3/2. 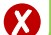 | To find the equation of the line parallel to the given vector passing through the given point, we can use the point-slope form of the equation of a line. The slope of this line is the same as the slope of the given vector, i.e., -1/2. Plugging in the coordinates of the given point (2, 1) and the slope into the point-slope form, we have: y - 1 = (-1/2)(x - 2) y - 1 = -1/2x + 1: y = -1/2x + 2 Therefore, the equation is y = -1/2x + 2. | In two dimensions, the equation of a line can be written in slope-intercept form: y = mx + b, where m is the slope and b is the y-intercept. In this case, the given vector is (−2, 1), so the slope of the line m is 1/-2 = -1/2. Since we know the line passes through the point (2, 1), we can substitute these values into the equation and solve for b (the y-intercept): 1 = (-1/2)x2 + b, 1 = -1 + b, b = 2. So, the equation of the line is y = -1/2x + 2. 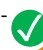 |

Figure 11: **Arithmetic Example.** DebateGPT-3.5 shows improvements in solving simple examples related to multiplication, addition, division and subtraction. DebateGPT-3.5 seems to have a better understanding of order-of-operations and PEMDAS.

**Prompt:** Can you answer the following question as accurately as possible? According to Nagel, the problem of moral luck arises because:: A) philosophers try to apply arbitrarily stringent standards to moral responsibility., B) the self is threatened with dissolution by the absorption of its acts into the class of events., C) our moral theories are hopelessly off track., D) our theories of human action are hopelessly off track.

| GPT-3.5: | DebateGPT-3.5: | GPT-4: |
|---|---|---|
| The correct answer is (D). Nagel argues that humans are hopeless off track since our moral judgments are based significantly on factors beyond an individual's control. This means that moral judgements can never match with our moral responsibility. 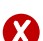 | The problem of moral luck arises because our theories of human action are hopelessly off track. Nagel argues that our moral judgments are based significantly on factors beyond an individual's control, like luck, which challenges the traditional view that individuals are only morally responsible for actions they intentionally choose. This problem reveals a discrepancy between our common-sense moral judgments and our philosophical theories of moral responsibility. (D) 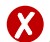 | According to philosopher Thomas Nagel, the problem of moral luck arises primarily because: (B) the self is threatened with dissolution by the absorption of its acts into the class of events. Nagel argues that moral luck is a problem because it challenges our sense of moral responsibility and agency. He suggests that our moral judgments are often influenced by factors beyond our control, such as the consequences of our actions, and this threatens the idea of a stable, morally responsible self. |

Figure 12: **Incorrect MMLU example.** GPT-3.5 and DebateGPT-3.5 incorrectly summarize the works of Nagel, likely due to lack of such examples in training. We notice that DebateGPT and GPT-3.5 give very similar explanations for their response, suggesting that this incorrect association may be due to specific halluncinations in GPT-3.5 that remained incorrect in DebateGPT-3.5.

**Prompt:** Given a set of points, calculate the area of the shape created. (1, 2) (2, 3) (3, 2) (4, 5)

**GPT-3.5:**

Wec an see that the two triangles formed are (1, 2) (2, 3) (3, 2) and (2, 3) (3, 2) (4, 5). Triangle 1: Base = distance between points (1, 2) and (3, 2) = 2 units Height = distance between point (2, 3) and the line containing the base = 1 unit (since the line containing the base is the horizontal line passing through (2, 3)) Therefore, Area of Triangle 1 = 1/2 x base x height = 1/2 x 2 x 1 = 1 square unit Triangle 2: Base = distance between points (2, 3) and (4, 5) = sqrt((4-2)^2 + (5-3)^2) = sqrt(8) units. Height = distance between point (3, 2) and the line containing the base = 2 units Therefore, Area of Triangle 2 = 1/2 x base x height = 1/2 x sqrt(8) x 2 = 2. Adding areas of both triangles, Area = 3 ❌

**DebateGPT-3.5:**

Area = 1/2 x base x height
Based on the coordinates of the points given, we can set up the following triangles and calculate their areas:
Triangle 1: (2-1)(3-2)/2 = 0.5
Triangle 2: (2-1)(5-2)/2 = 1.5
Triangle 3: (3-2)(5-2)/2 = 1.5
Triangle 4: (4-2)(5-2)/2 = 3
To calculate the area of the shape created, we need to subtract the areas of the smaller triangles from the area of the larger triangle formed by connecting points 1, 2, and 4:
Area = (4-1)(5-2)/2 - (sum of areas of the smaller triangles)
Area = 0.5 ❌

**GPT-4:**

Assuming that they form a polygon in order (i.e., (1,2) to (2,3) to (3,2) to (4,5) and back to (1,2)), we could apply the Shoelace formula—a method for computing the area of a polygon whose vertices are described by Cartesian coordinates.
The Shoelace formula is [(Sum of x[i]y[i+1]) - (Sum of y[i]x[i+1])] / 2
The formula applied will be
= 1/2 [ (1x3 + 2x2 + 3x5 + 4x2) - (2x2 + 3x3 + 2x4 + 5x1) ]
= 1/2 [ (19) - (20) ]
= 1/2 x (1)
= 0.5 ❌

Figure 13: **Incorrect Alpaca Example**. We see an example where the goal is to find the area of a polygon represented by Cartesian coordinates which all three models get incorrect. We observe that GPT-3.5 and DebateGPT-3.5 use decomposition into smaller polygons while GPT-4 correctly attempts to use the Shoelace formula but obtains the incorrect answer. Such gaps may be overcome through larger training datasets.

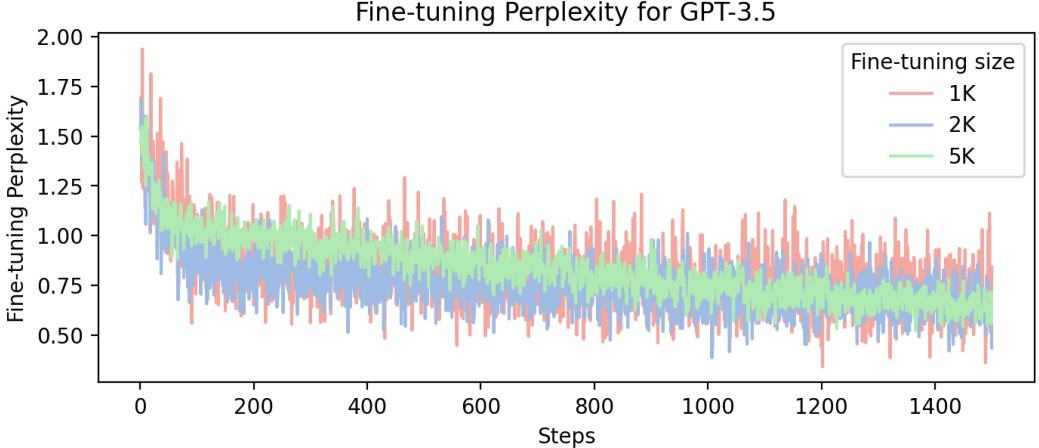

Figure 14: **DebateGPT-3.5 Fine-tuning Perplexity**. Fine-tuning perplexity for different dataset sizes while fine-tuning GPT-3.5 for 10 epochs. Each epoch consists of 150 update steps while fine-tuning. We see that for each dataset size, GPT-3.5 can fit to our data and perplexity consistently decreases.

| Model | Confidence Faithfulness (%) ↑ |
|---|---|
| First Round | 63.00 |
| Last Round | 82.00 |

Table 8: **Confidence Scoring Assessment**. We assess the quality of confidence scoring during multi-agent debate using 100 Alpaca examples from our fine-tuning dataset. We analyze the responses to see whether the confidence score accurately reflects the correctness/quality of the response i.e. does a higher confidence score refer to a more correct/high quality response and use a binary yes/no answer. We report the percentage of yes responses in our human evaluation from the first round responses and last round responses. We see an increase in the faithfulness of our confidence scores in the last round of debate compared to the first round.

| Model | MMLU Accuracy - 100 examples (%) ↑ |
|---|---|
| GPT3.5 | 62.00 |
| DebateGPT-3.5 Full Debate | 68.00 |
| DebateGPT-3.5 (Ours) | 73.00 |

Table 9: **Multi-agent Debate Fine-tuning Dataset Construction** We present results on 100 MMLU examples comparing GPT-3.5 with a DebateGPT-3.5 constructed by fine-tuning on the entire debate demonstration (*DebateGPT-3.5 Full Debate*) and with our method of fine-tuning DebateGPT-3.5 on the final round response only. We see that DebateGPT-3.5 fine-tuned with only the final round response is consistently better when evaluated on 100 MMLU examples.

