# OpenReview forum: "DebateGPT: Fine-tuning Large Language Models with Multi-agent Debate Supervision"
_ICLR.cc/2024/Conference — Submitted to ICLR 2024_

### Official Review · Reviewer_9fkA · 2023-10-28

**Soundness:** 2 fair
**Presentation:** 3 good
**Contribution:** 2 fair
**Rating:** 3
**Confidence:** 4

**Summary:**

The paper proposes a method called DebateGPT, which uses data generated through multi-agent debate to  fine-tune GPT-3.5. The authors show that DebateGPT achieves comparable performance to GPT-4 on various tasks, including instruction following, commonsense reasoning, and mathematics.

**Strengths:**

1. The proposed method achieves impressive results, with DebateGPT showing comparable performance to GPT-4.
2. The paper provides detailed explanations and analysis of the proposed method.

**Weaknesses:**

1. If I'm understanding correctly, the Multi-Agent debate method used to generate data seems to following the approach in [1]. However, it's important to note that when comparing it to the baseline models, we're only looking at GPT-3.5 and GPT-4, and there's no direct mention of [1].
2. In the main contributions of the paper, it claims to "offers a more economical alternative to leveraging models like GPT-4". However, when we look at Figure 6, we see that both options are similarly priced. Furthermore, taking into account their divergent performance makes it difficult for me to fully support this claim.

[1] Improving Factuality and Reasoning in Language Models through Multiagent Debate. https://arxiv.org/pdf/2305.14325.pdf

**Questions:**

1. Which LLMs are included in the Multi-Agent used for data generation?

---

> ### Author Response · Authors · 2023-11-23
> **Response to reviewer 9fkA (1/1)**
>
> Thank you for your thoughtful review! We respond to the points below:
>
> **Question:If I'm understanding correctly, the Multi-Agent debate method used to generate data seems to following the approach in [1]. However, it's important to note that when comparing it to the baseline models, we're only looking at GPT-3.5 and GPT-4, and there's no direct mention of [1].**
>
> **Answer**: We thank the reviewer for the valuable comment. We have added the results of fine-tuning GPT3.5 using the data generated by the method proposed by [1] on three evaluation benchmarks. Our method consistently outperforms [1].
> | Model | AlpacaEval | Arithmetic | Winogrande |
> | -------- | --------------- | -------------- | ---------------- |
> | DebateGPT-3.5 Original Debate[1] | 85.64 | 50.30 | 84.19 |
> | DebateGPT-3.5 (Ours) | 91.91 | 55.30 | 86.97 |
>
> **Question:In the main contributions of the paper, it claims to "offers a more economical alternative to leveraging models like GPT-4". However, when we look at Figure 6, we see that both options are similarly priced. Furthermore, taking into account their divergent performance makes it difficult for me to fully support this claim.**
>
> **Answer**: Figure 6 shows that on 500 examples, generating responses to 500 examples with multi-agent debate costs 30 dollars less than GPT-4. As the number of examples increases, this cost will add up considerably. In our case with generating a dataset of 5K examples, we saved approximately 300 dollars using our approach with multi-agent debate.
>
> Furthermore, our performance is comparable with GPT-4. Table 1 shows that our method has a win rate accuracy of 88.6 and GPT-4 is 89.0. Table 6 in Appendix F (pasted below) shows that our method is consistently better than other methods at generating high quality data.
>
> | Model                            	| Win Rate (higher is better) |
> |--------------------------------------|----------|
> | GPT-3.5                          	| 74.2 	|
> | Zero-Shot Chain of Thought       	| 78.4 	|
> | Multi-Agent Debate (Du et. al. 2023) | 77.2 	|
> | Multi-Agent Debate (Ours)        	| 81.2 	|
>
>
> **Question: Which LLMs are included in the Multi-Agent used for data generation?**
>
> **Answer**: As stated in Section 3.2, we use GPT-3.5 in the multi-agent debate for data generation.

---

### Official Review · Reviewer_8M4v · 2023-10-28

**Soundness:** 2 fair
**Presentation:** 3 good
**Contribution:** 2 fair
**Rating:** 5
**Confidence:** 3

**Summary:**

The paper introduces DebateGPT, which fine-tuned GPT-3.5 using a dataset generated through multi-agent debate, reducing the necessity for resource-intensive human intervention. By augmenting this method with techniques such as summarization, confidence scoring, and data refinement, the dataset's quality sees significant enhancement. DebateGPT demonstrates performance levels on par with GPT-4 across a diverse range of tasks, including domains like commonsense reasoning and mathematics. Notably, DebateGPT's distinguishing feature lies in its relatively compact size compared to GPT-4, all while relying on a modest dataset, thereby challenging the prevailing notion that larger models are heavily reliant on extensive human feedback.

**Strengths:**

They employ a multi-agent debate approach to generate the dataset, and through the incorporation of elements like confidence scoring, summarization, and data cleaning, they demonstrate a noticeable enhancement in dataset quality when compared to the GPT-3.5 baseline.

**Weaknesses:**

The primary contribution of this paper lies in its application of multi-agent debate for dataset creation. The experiments on dataset quality reveal that the introduction of multi-agent debate, along with summarization, confidence scoring, and data cleaning, results in an improved dataset quality. However, it's worth noting that the utilization of confidence scores and filtering techniques is a standard practice in the field of dataset augmentation, lacking any groundbreaking innovation apart from the incorporation of multi-agent debate. To make a meaningful assessment of its effectiveness, more comprehensive experiments should be conducted to compare these approaches with existing methods in data augmentation.

Furthermore, the paper exclusively reports fine-tuned results on the dataset it generated. It remains unclear how fine-tuning the model on baseline datasets would perform in comparison. This omission prevents a thorough understanding of the extent to which a dataset with better quality can impact downstream tasks.

Another important consideration is the potential impact of fine-tuning on the model's generalization abilities for tasks beyond those presented in the paper. This aspect warrants further investigation to ascertain the overall implications of the fine-tuning process.

**Questions:**

N/A

---

> ### Author Response · Authors · 2023-11-23
> **Response to reviewer 8M4v (1/1)**
>
> Thank you for your thoughtful review! We respond to the points below:
>
> **Question: it's worth noting that the utilization of confidence scores and filtering techniques is a standard practice in the field of dataset augmentation, lacking any groundbreaking innovation apart from the incorporation of multi-agent debate.**
>
> **Answer**: Based on our knowledge, confidence scores and filtering techniques are the first time used for data generation under the multi-agent debate framework. We haven’t seen any work use confidence scores or filtering specifically in designing text datasets. We would appreciate it if the reviewer could point to papers that use confidence scores and filtering techniques in the field of dataset augmentation.
>
> **Question: To make a meaningful assessment of its effectiveness, more comprehensive experiments should be conducted to compare these approaches with existing methods in data augmentation.**
>
> **Answer**: We thank the reviewer for the valuable comment. We include more baseline methods for generating data and measure the data quality of each method in comparison with our design with multi-agent debate on 1000 examples randomly sampled from Alpaca. We report this in Appendix F. We see that our method consistently generates higher quality data than every other prompting method. We also include this table here:
>
> | Model                            	| Win Rate (higher is better) |
> |--------------------------------------|----------|
> | GPT-3.5                          	| 74.2 	|
> | Zero-Shot Chain of Thought       	| 78.4 	|
> | Multi-Agent Debate (Du et. al. 2023) | 77.2 	|
> | Multi-Agent Debate (Ours)        	| 81.2 	|

---

### Official Review · Reviewer_x4r2 · 2023-11-02

**Soundness:** 3 good
**Presentation:** 3 good
**Contribution:** 2 fair
**Rating:** 5
**Confidence:** 4

**Summary:**

Pros:
1. The author proposes a novel framework that utilizes multi-agent debate for data generation and supervised training. Multiple interesting features could be seen such as the confidence score, summarization, and cleaning.
2. The results presented in the paper show that the model has increased performance on various benchmarks including commonsense reasoning, mathematics, factuality, etc.

Cons:
1. The supervised data seems composed of questions and answers given by the multi-debate framework, rather than multi-agent debate demonstrations. It does not make so much sense to me as there are incorrect answers in the answers as well, right? (Plz correct me if any falsehood) Have the authors tried using answers given by human annotators?
2. The author claims that the DebateGPT-3.5 is an economical alternative for GPT4, which could easily regarded as over-hyping (I am NOT doubting the results). However, the author should present more enriched results on more benchmarks.

Questions:
1. Human-in-loop is so expensive,  utilizing other methods including debate might be a panacea, but need to carefully investigate. Does the multi-agent debate framework apply to the evaluation in the experiments?
2. The confidence score sounds interesting, however, the reliability of the scalar given by LLM is still worth doubting (an empirical concern from the reviewer), has the author done any faithfulness investigation on that?
3. The summarization makes sense to me, however, the answers from other agents will be compressed into a short answer, which inevitably brings a loss of information, has the author tried to handle that problem?
4. agents only debate one time before issuing the final answer. How could they possibly converge to an answer with only ONE iteration? It sounds a little bit brutal to simply generalize the results before multi-iteration convergence (Du et al., 2023).

**Strengths:**

See summary

**Weaknesses:**

See summary

**Questions:**

See summary

---

> ### Author Response · Authors · 2023-11-23
> **Response to reviewer x4r2 (1/2)**
>
> Thank you for your thoughtful review! We respond to your points below.
>
> **Question: The supervised data seems composed of questions and answers given by the multi-debate framework, rather than multi-agent debate demonstrations. It does not make so much sense to me as there are incorrect answers in the answers as well, right? (Plz correct me if any falsehood) Have the authors tried using answers given by human annotators?**
>
> **Answer**: This is correct. We construct our dataset by extracting the question and the final round response since our findings show that responses in later rounds are more accurate than responses in earlier rounds of multi-agent debate.
> We show the results comparing DebateGPT-3.5 with a GPT-3.5 fine-tuned on the entire debate demonstration. We randomly select 100 examples from two evaluation datasets, GSM and MMLU, respectively. In the table below, we see that that fine-tuning on the final response is superior to fine-tuning on the entire debate demonstration. This is because when we fine-tune using the entire debate demonstration, the fine-tuned model is confused by the incorrect information in the early rounds of debate. Another reason is that the length of some demonstrations exceeds the context window length of GPT-3.5 and the conversations in later rounds of debate are discarded. Thus the method we proposed in the paper that utilizes the most accurate final debate results achieves better performance.
>
> The results have been added to Appendix H of the paper and are included below.
>
> | Model | MMLU |
> | -------- | ------- |
> | GPT-3.5 | 62.00 |
> | DebateGPT-3.5 (Full Debate) | 68.00 |
> | DebateGPT-3.5 (Ours) | 73.00 |
>
> Multi-agent debate can generate incorrect answers like any automatic generation method. However, our goal in this work is to avoid human annotation, which is expensive, time-consuming, and hard to scale, while constructing datasets (See paragraph 1 of our introduction in our paper) . Based on this, our paper shows that multi-agent debate can generate high quality data in an efficient manner and this high quality data can lead to stronger fine-tuning results as shown in the paper and the results above.
>
> **Question: Human-in-loop is so expensive, utilizing other methods including debate might be a panacea, but need to carefully investigate. Does the multi-agent debate framework apply to the evaluation in the experiments?**
>
> **Answer**: We assume the reviewer is asking whether multi-agent debate can be added as an evaluation baseline. To be clear, multi-agent debate is only used for data generation in our paper. Our evaluation only uses a single agent the same as baseline methods for fair comparisons. However, multi-agent debate can also be used for more accurate evaluation which is not the focus of this paper.
>
>
> **Question: The confidence score sounds interesting, however, the reliability of the scalar given by LLM is still worth doubting (an empirical concern from the reviewer), has the author done any faithfulness investigation on that?**
>
> **Answer**: This is a good point! We have conducted a preliminary investigation into how reliable the confidence score given by the LLM is. Based on our results, LLMs usually have a confidence score > 90 and are even confident in incorrect answers. However, in multi-agent debate, we find that the agents will adjust their response based on the response from other agents, and responses with higher confidence scores are more likely to be accepted by the other agents.
>
> To better understand the reliability of our confidence score, we design a small human evaluation. We sample 100 examples from our dataset. For each example, we had the evaluator analyze the quality of the response. The evaluator then analyzed the confidence score in the final round. If the confidence score was low (< 95) and the quality of the response was low, or if the confidence score was high (>= 95) and the quality of the response was high, then we say that the confidence score reliably reflects the quality of the response. We calculated the percentage of examples where the confidence score reliably reflects the response quality according to our definition in the 100 examples. We find 82\% of confidence scores in the last round of debate accurately reflected the correctness.  Our human evaluation shows that the confidence scoring method in our paper can reflect the relative accuracy of the generated response, which is reliable enough as a reference when agents participate in multiagent debate.

---

> > ### Author Response · Authors · 2023-11-23
> > **Response to reviewer x4r2 (2/2)**
> >
> > **Question: The summarization makes sense to me, however, the answers from other agents will be compressed into a short answer, which inevitably brings a loss of information, has the author tried to handle that problem?**
> >
> > **Answer**: We agree with the reviewer. We do handle this problem in our paper. We tried many different summarization prompt designs and we chose the prompt to prevent information losswhile removing the redundant information. We include two example prompts here and show the kind of summarization we obtained.
> >
> > Given a question from the MMLU dataset as an example:
> >
> > **Question Prompt**: Can you answer the following question as accurately as possible? The intake of which minerals is not affected by high intakes of foods containing phytic acid? : A) Calcium, B) Potassium, C) Zinc, D) Iron
> >
> > We show two summarization prompts we tested:
> >
> > **Summarization Prompt 1**: Summarize the responses from other agents into a short sentence
> >
> > **Model Response**: “One agent believes that high intakes of foods containing phytic acid can affect the absorption of minerals like calcium, potassium, zinc (to a lesser extent) chooses zinc (C), while the other agent states iron (D) is the most strongly inhibited by phytic acid.”
> >
> > **Summarization prompt 2**: Summarize the responses from each agent in one response.
> >
> > **Model Response**: “Different agents provided different responses to the question of which mineral intake is not affected by high intakes of foods containing phytic acid. One agent stated that potassium (B) is not affected by phytic acid as it is not bound or affected by it. However, one agent stated that zinc (C) intake is not affected by phytic acid due to its ability to form soluble complexes with phytic acid, making it readily absorbable by the body. Overall, there is a consensus that calcium (A) and iron (D) intake may be affected by high intakes of foods containing phytic acid.”
> >
> > When we compare the model response to prompt 1 and prompt 2, we see prompt 1, which asks for a one sentence summary, causes information loss. Prompt 2 succinctly combines the responses into one response, removing redundant information and keeping relevant information such as reference to the confidence score.
> >
> > We see this reasoning reflected in the data quality. Information loss reduces Alpaca data quality as indicated below.
> >
> > | Summarization Prompt | Win Rate (%) |
> > | -------------------------------| ------------------ |
> > | Summarize the responses from other agents into a short sentence.  | 65 |
> > | Summarize the responses from each agent in one response. | 79 |
> >
> > We include these results in Appendix E of the paper. These results indicate that our summarization prompt removes redundant information without removing important information and improves data quality.
> >
> >
> > **Question: agents only debate one time before issuing the final answer. How could they possibly converge to an answer with only ONE iteration? It sounds a little bit brutal to simply generalize the results before multi-iteration convergence (Du et al., 2023).**
> >
> > **Answer**: Just to clarify, agents debate over multiple rounds -- so not one time. In fact, we find that increasing the number of rounds improves results. In our paper (Section 4.3), we use multiple agents and multiple rounds of debate, specifically 4 agents and 3 rounds.

---

### Public Comment · ~Guohao_Li1 · 2023-11-14
**Suggesting related work**

The paper introduces "DebateGPT," a large language model (LLM) fine-tuned using a novel method called multi-agent debate supervision. This method fine-tunes GPT-3.5 using responses generated from debates among multiple less robust LLMs, without requiring human annotations. DebateGPT demonstrates comparable performance with GPT-4 on various tasks, particularly excelling in zero-shot generalization to new tasks like commonsense reasoning, factuality, and mathematics. Key innovations include confidence scoring, summarization, and cleaning steps in the multi-agent debate process, making it a cost-effective alternative to using more resource-intensive models like GPT-4.

Thanks for the great work. It could also be beneficial to discuss prior work on using multi-LLM agent cooperation to generate instruction finetuning datasets [1].

[1] Li, Guohao, Hasan Abed Al Kader Hammoud, Hani Itani, Dmitrii Khizbullin, and Bernard Ghanem. "CAMEL: Communicative Agents for" Mind" Exploration of Large Language Model Society." NeurIPS 2023

---

> ### Author Response · Authors · 2023-11-23
> **Related Work Suggestion**
>
> **Question: It could also be beneficial to discuss prior work on using multi-LLM agent cooperation to generate instruction finetuning datasets [1].**
>
> **Answer**: Thank you for your interest in our paper! We have added discussion of [1] to our related work section, highlighted in blue!
>
> [1] Li, Guohao, Hasan Abed Al Kader Hammoud, Hani Itani, Dmitrii Khizbullin, and Bernard Ghanem. "CAMEL: Communicative Agents for" Mind" Exploration of Large Language Model Society." NeurIPS 2023

---

> > ### Public Comment · ~Guohao_Li1 · 2023-11-26
> >
> > Thank you for sharing your valuable insights in your paper! The exploration of multi-LLM agent cooperation and debate supervision is indeed an intriguing direction. Regarding your recent revision, I noticed that reference [1] appears to be missing. This could be an oversight, and I kindly recommend considering its inclusion in the final version, if it aligns with your perspective.

---

### Author Response · Authors · 2023-11-23
**General Response (1/1)**

We thank the reviewers for their thorough feedback. We are glad Reviewers x4r2 and 8M4v found our framework of generating datasets using multi-agent debate supervision novel and all three reviewers found our improvements over GPT-3.5 substantial. We cover some changes to the paper below:

1. As suggested by reviewer x4r2, we introduce appendix sections to analyze other aspects of our approach including confidence scores, the use of summarization and the summarization prompt, and fine-tuning using the debate demonstration.
2. As suggested by both reviewer 8M4v and reviewer 9fKA, we investigate and benchmark other methods of generating data such as CoT or the originally proposed multi-agent methodology [1], showing that our version of multi-agent debate consistently generates higher quality data.
3. As suggested by reviewer 9fKA we include a new evaluation baseline the fine-tuned model performance using data generated by our method and data generated by the originally proposed multi-agent debate [1].
Changes in the paper are highlighted in blue. Please refer to our detailed response below for further modifications based on the feedback provided by the reviewers.

[1] Du et. al. Improving Factuality and Reasoning in Language Models through Multiagent Debate. https://arxiv.org/pdf/2305.14325.pdf

---

### Meta-Review · Area_Chair_jusA · 2023-12-07

**Metareview:**

This paper leverages multi-agent debate framework to generate finetuning data (from Alphaca), which are used to finetune GPT-3.5 models. After reviewing the feedback from reviewers and the author response, I also take a close look at the paper. The major weakness of the paper is its lack of sufficient novel contribution. The key idea is built upon the method developed in the paper "Improving Factuality and Reasoning in Language Models through Multiagent Debate" (https://arxiv.org/pdf/2305.14325.pdf) to generate the finetuning data with minor modifications such as adding confidence scores. The main difference is that it uses the generated data to finetune the model, which is a relatively straightforward idea.

**Justification For Why Not Higher Score:**

The key reason is that this paper does not have sufficient novelty and the results are not convincing and impressive.

**Justification For Why Not Lower Score:**

N/A

---

### Decision · Program_Chairs · 2024-01-16

Reject